# Dasatinib and Quercetin Alleviate Retinal Ganglion Cell Dendritic Shrinkage and Promote Axonal Regeneration in Mice with Optic Nerve Injury

**DOI:** 10.3390/ijms262412170

**Published:** 2025-12-18

**Authors:** Xin Bin, Shuyi Zhou, Yanxuan Xu, Si Chen, Shaowan Chen, Wen Yao, Yingjie Cao, Kunliang Qiu, Tsz Kin Ng

**Affiliations:** 1Joint Shantou International Eye Center of Shantou University and the Chinese University of Hong Kong, Shantou 515041, China; 2Department of Ophthalmology and Visual Sciences, The Chinese University of Hong Kong, Hong Kong, China

**Keywords:** optic nerve injury, retinal ganglion cells, dendrites, axons, dasatinib, quercetin

## Abstract

Optic nerve (ON) injury by trauma induces progressive retinal ganglion cell (RGC) death and axonal loss, which leads to irreversible visual impairment and even blindness. Recently, we discovered that cellular senescence is involved in RGC survival regulation post-ON injury, and senolytic (dasatinib and quercetin) treatments can promote RGC survival and electroretinography activity. Here, we aimed to further evaluate the effects of dasatinib and quercetin on RGC dendrites and axons in mice with an ON crush injury. Longitudinal in vivo imaging analysis demonstrated that the RGC dendritic shrinkage was significantly reduced in mice with both individual and combined treatment of dasatinib and quercetin as compared to the vehicle treatment group. Similarly, dasatinib and quercetin treatments significantly promoted axonal regeneration post-ON injury as compared to the vehicle-treated mice. RNA sequencing analysis showed that the differentially expressed genes were enriched in the response to glucocorticoid, calcium ion binding, and cerebral cortex development. Sybr green PCR and immunofluorescence analyses validated that the axonal extension-related gene, meteorin (*Metrn*), was significantly upregulated in the dasatinib-only and combined dasatinib and quercetin treatments. In summary, this study revealed that dasatinib and quercetin alleviated RGC dendritic shrinkage and promoted axonal regeneration in mice after ON injury, probably mediated through meteorin, suggesting the dendrite repair and axonal regeneration potentials of dasatinib and quercetin for traumatic optic neuropathy treatment.

## 1. Introduction

Traumatic optic neuropathy refers to the acute injury to the optic nerve (ON) due to trauma, which can lead to irreversible visual impairment and even blindness [1]. The World Health Organization estimated that trauma causes 2.3 million individuals with binocular low vision, 19 million with monocular blindness, and 1.6 million with binocular blindness worldwide [2]. Progressive retinal ganglion cell (RGC) death and axonal loss are major causes of visual field defects post-ON injury [3]. As a central nervous system (CNS) neuron, RGCs show limited intrinsic regenerative ability. In addition, there is still no effective clinical treatment for reducing RGC dendritic shrinkage and promoting axonal regeneration after ON injury. Therefore, exploring new treatments targeting RGC dendrites and axons in cases of ON injury is warranted.

Multiple strategies have been reported to promote RGC axonal regeneration in rodents after ON injury, including the deletion of *Klf4* [4] and *Pten*/*Socs3* genes [5], the inhibition of non-muscle myosin II [6], and the application of stromal cell-derived factor 1 [7]. Moreover, our previous studies also demonstrated that the administration of mesenchymal stem cells and green tea extract can promote RGC survival and axonal regeneration in rats post-ON injury [8,9,10]. Recently, we discovered the involvement of cellular senescence in the regulation of RGC survival in mice post-ON injury, and we applied the treatments of senolytics (dasatinib and quercetin) to confirm cellular senescence-mediated RGC death regulation post-ON injury and to verify the effect of transforming growth factor-β receptor inhibitor treatment on RGC survival promotion [11]. Dasatinib has been reported to protect RGCs from apoptosis in mice with ischemia injury [12]. Similarly, quercetin was also studied to improve the survival of RGCs in glaucoma and ischemia rat models [13,14]. Despite having demonstrated that dasatinib and quercetin treatments can promote RGC survival and enhance the electroretinography activity in our previous study [11], the effects of dasatinib and quercetin on RGC dendrites and axons in the ON injury model have still not been investigated. Herein, in addition to the RGC survival, we aimed to further evaluate the effects of dasatinib and quercetin on RGC dendrites and axons in mice with ON crush injury. In addition, the potential underlying mechanism was also investigated by transcriptomic analysis.

## 2. Results

### 2.1. Dasatinib and Quercetin Treatments Alleviate Retinal Ganglion Cell Dendritic Shrinkage in Mice Post-Optic Nerve Injury

The dendritic morphology of RGCs was assessed longitudinally in the B6.Cg-Tg(Thy1-YFP)HJrs/J (Thy1-YFP-H) mice before and after ON injury by confocal scanning laser ophthalmoscope (Figure 1A). The dendritic analysis showed that RGCs with dasatinib treatment alone showed a significantly larger area of dendritic field (0.055 ± 0.017 mm^2^, *p* = 0.002; Figure 1B), longer total dendritic branch length (1.72 ± 0.57 mm, *p* = 0.013; Figure 1C), more dendritic terminal branches (13.54 ± 2.11, *p* = 0.022; Figure 1D), and higher dendritic branching complexities (148.38 ± 47.02, *p* = 0.012; Figure 1E) as compared to the vehicle control group at Day 14 post-ON injury (area of dendritic field: 0.034 ± 0.011 mm^2^; total dendritic branch length: 1.17 ± 0.38 mm; dendritic terminal branches: 10.80 ± 2.90; dendritic branching complexities: 101.10 ± 33.80). Similarly, RGCs with a combined dasatinib and quercetin treatment also showed significantly larger area of dendritic field (0.048 ± 0.017 mm^2^, *p* = 0.033), longer total dendritic branch length (1.69 ± 0.50 mm, *p* = 0.014), more dendritic terminal branches (13.40 ± 3.42, *p* = 0.024), higher dendritic branching complexities (147.47 ± 43.17, *p* = 0.011), and higher maximum Sholl score (12.93 ± 2.79, *p* = 0.019; Figure 1F) as compared to the vehicle control group at Day 14 post-ON injury (maximum Sholl score: 10.50 ± 2.59). In contrast, RGCs with quercetin treatment alone just showed significantly more dendritic terminal branches (14.80 ± 2.34, *p* = 0.001) and higher maximum Sholl score (13.60 ± 2.26, *p* = 0.003) as compared to the vehicle control group at Day 14 post-ON injury. Collectively, our results indicated that individual and combined treatment of dasatinib and quercetin can significantly alleviate RGC dendritic shrinkage in mice after ON injury.

### 2.2. Dasatinib and Quercetin Treatments Promote Retinal-Ganglion-Cell Axonal Regeneration in Mice Post-Optic Nerve Injury

Next, we investigated the treatment effects of dasatinib and quercetin on RGC axonal regeneration in mice with ON injury in vivo. The immunofluorescence analysis demonstrated that the number of regenerated axons in the dasatinib-treated mice (250 μm: 61.05 ± 13.26 axons/ON, *p* < 0.001; 500 μm: 10.87 ± 4.35 axons/ON, *p* = 0.001) was significantly higher than that in the vehicle-treated mice (250 μm: 7.72 ± 2.61 axons/ON, 500 μm: 1.31 ± 1.81 axons/ON) at 14-day post-ON injury (Figure 2). Similarly, the number of regenerated axons in the quercetin treatment group (250 μm: 69.81 ± 10.88 axons/ON, *p* < 0.001; 500 μm: 17.72 ± 5.61 axons/ON, *p* < 0.001) was also significantly higher than that in the vehicle control group at 14-day post-ON injury. In addition, significantly more regenerated axons were observed in the mice with combined treatment of dasatinib and quercetin (250 μm: 73.82 ± 15.49 axons/ON, *p* < 0.001; 500 μm: 22.17 ± 3.89 axons/ON, *p* < 0.001) as compared to the vehicle control, and it was also significantly higher than the dasatinib treatment group (500 μm: *p* < 0.001). Our results indicated that individual and combined treatment of dasatinib and quercetin can significantly promote RGC axonal regeneration in mice after ON injury.

### 2.3. Retinal Transcriptomic Analysis of Dasatinib and Quercetin Treatments in Mice Post-Optic Nerve Injury

To delineate the mechanisms of dasatinib and quercetin treatments for dendritic repair and axonal regeneration, we performed RNA sequencing analysis on the retinas of mice with combined treatment of dasatinib and quercetin at Day 5 post-ON injury. Principal component analysis (Figure 3A) and hierarchical clustering analysis (Figure 3B) confirmed that the gene expression profiles of the retinas of the mice with the combined treatment of dasatinib and quercetin and those of the vehicle treatment were differentially clustered. In total, 25,131 genes were identified. Among which, 43 genes were differentially expressed in the retina of the mice with the combined treatment of dasatinib and quercetin as compared to that of the vehicle treatment (Figure 3C), among which 5 protein-coding genes (*Rpl19*, *Pdc*, *Nrtn*, *Metrn*, and *Tma7*) were upregulated and 10 protein-coding genes (*Fstl5*, *Uba6*, *Slc38a2*, *Zfhx4*, *Ryr3*, *Pcsk1*, *Fosl2*, *Fat4*, *Papolb*, and *Mthfd2*) were downregulated (Table 1). The gene ontology analysis showed that the 15 differentially expressed protein-coding genes were involved in cerebral cortex development (*p* = 0.043), calcium ion binding (*p* = 0.041), and response to glucocorticoid (*p* = 0.027) (Table 2).

### 2.4. Validation of Gene Expression in Dasatinib- and Quercetin-Treated Mice Post-Optic Nerve Injury

To verify the results of the RNA sequencing analysis, all 15 differentially expressed protein-coding genes were subjected to SYBR Green PCR verification analysis on the retinas of mice treated with dasatinib alone, quercetin alone, and a combination of dasatinib and quercetin at Day 5 post-ON injury. The SYBR Green PCR analysis demonstrated that the expression of the meteorin (*Metrn*) gene was significantly upregulated in the dasatinib-treatment-alone group (2.64 ± 0.96 folds, *p* < 0.001) and the combined dasatinib and quercetin treatment group (2.08 ± 0.20 folds, *p* = 0.009) as compared to that of the vehicle treatment group (Figure 4). Moreover, the expressions of the *Slc38a2* and *Mthfd2* genes were significantly downregulated in the quercetin-treatment-alone group by 50.46% (*p* = 0.001) and 37.71% (*p* = 0.047), respectively, as compared to those of the vehicle treatment group. In contrast, the remaining examined genes (*Fat4*, *Fosl2*, *Fstl5*, *Nrtn*, *Papolb*, *Pcsk1*, *Pdc*, *Rpl19*, *Ryr3*, *Tma7*, *Uba6*, and *Zfhx4*) did not show statistically significant differences in the dasatinib-treatment-alone group, quercetin-treatment-alone group, and combined dasatinib and quercetin treatment group as compared to that of the vehicle treatment group (*p* > 0.05).

### 2.5. Expression Analysis of Meteorin in the Retina of Dasatinib- and Quercetin-Treated Mice Post-Optic Nerve Injury

To further validate the *Metrn* gene expression upon dasatinib and quercetin treatments, we examined the protein expression of meteorin in the retinal sections of dasatinib- and quercetin-treated mice post-ON injury by immunofluorescence analysis. The measurements in the ganglion cell complex (GCC) layer demonstrated that the intensities of immunofluorescence signals of meteorin were significantly higher in the dasatinib-treatment-alone group (142.26 ± 46.82, *p* < 0.001) and the combined dasatinib and quercetin treatment group (100.12 ± 22.16, *p* = 0.001) as compared to that of the vehicle treatment group (62.69 ± 7.61; Figure 5). In contrast, the immunofluorescence signal intensity in the GCC layer of the quercetin-treatment-alone group (75.05 ± 21.22) showed no statistically significant difference as compared to that of the vehicle treatment group (*p* = 0.176). Our results indicated that meteorin could be involved in the dendritic repair and axonal regeneration of the dasatinib treatment.

## 3. Discussion

Results from this study showed the following: (1) dasatinib and quercetin treatments can alleviate RGC dendritic shrinkage in mice after ON injury; (2) dasatinib and quercetin treatments can promote RGC axonal regeneration in mice after ON injury; (3) the differentially expressed protein-coding genes in the mouse retina with combined treatment of dasatinib and quercetin were involved in the response to glucocorticoid, calcium ion binding, and cerebral cortex development; (4) the meteorin gene and protein were significantly upregulated in the treatments with dasatinib. Collectively, this study demonstrated the RGC dendritic repair and axonal regeneration abilities and mechanisms of dasatinib and quercetin in mice with ON injury.

Dasatinib and quercetin, as the commonly used senolytics, have long been applied to remove senescent cells [15,16]. Dasatinib was initially discovered as an ABL kinase inhibitor to treat leukemia [17], while quercetin belongs to the group of flavonoids with anti-tumor, anti-inflammatory, immunomodulatory, and antioxidant properties [18]. Dasatinib can penetrate into the central nervous system [19], and has been reported to protect RGCs from apoptosis in mice with ischemia injury [12], while quercetin has been shown to improve RGC survival in glaucoma and ischemia rat models [13,14], mitigate neuronal death in 6-hydroxydopamine-lesioned rats of Parkinson’s disease [20], and alleviate monosodium glutamate-induced excitotoxicity of spinal cord motoneurons in aged rats [21]. Combined treatment of dasatinib and quercetin has been shown to improve cognitive abilities in aged male Wistar rats [22], reduces seizure frequency in the *Mtor*^S2215F^ focal cortical dysplasia type II mouse model [23], alleviate over-training-induced learning and memory deficits in a rat hippocampus [24], and attenuate neurodegeneration and enhance neuron number in the ipsilateral cortex, hippocampus, and lateral posterior thalamus after traumatic brain injury [25]. Consistently, in our previous study, we have demonstrated that individual and combined treatment of dasatinib and quercetin can promote RGC survival and electroretinography activity and reduce cell-death-marker expression in mice post-ON injury [11], confirming the involvement of cellular senescence in RGC death regulation after ON injury and verifying the effect of transforming growth factor-β receptor inhibitor (LY2109761) treatment on RGC survival promotion. In this study, we further identified that the individual and combined treatment of dasatinib and quercetin can alleviate RGC dendritic shrinkage and promote axonal regeneration in mice after ON injury (Figure 1 and Figure 2). It has been reported that quercetin treatment can increase the neurite length of N1E-115 cells [26] and promote neurite outgrowth and increase the complexity of the neuronal branching trees of PC12 cells in vitro [27]. Moreover, quercetin can also promote axonal regeneration in rats after spinal cord injury [28] and enhance axonal regeneration of motoneurons in rats after spinal root avulsion and re-implantation [29]. In addition, the combined treatment of dasatinib and quercetin is associated with changes in the dendritic spine morphology of the apical dendritic tree of rat hippocampal CA1 neurons [22] and rescues the dendritic damage of the hippocampal neurons in mice with global cerebral ischemic brain injury [30]. Collectively, our results, together with other studies, suggest that dasatinib and quercetin show potential to promote neuronal survival and enhance dendritic repair and axonal regeneration against CNS disease/injury.

The combined treatment of dasatinib and quercetin has been shown to inhibit NF-κB activity in HEI-OC1 auditory cells and downregulate the inflammatory cytokines and chemokines in cochlear explants [31]. The combined treatment of dasatinib and quercetin can also reduce the apoptotic cells in rat hippocampus [24], and enhance mitochondrial cytochrome c oxidase activity and ATP production, reduce reactive oxygen species accumulation, and suppress oxidative damage to cellular protein structure in mouse hippocampal neurons [30]. Furthermore, the combined treatment of dasatinib and quercetin alleviated inflammation and changed the histone H3 methylation profile in aged male Wistar rats [22]. In this study, our results showed that the differentially expressed genes in the retinas of mice with combined treatment of dasatinib and quercetin were involved in the response to glucocorticoid, calcium ion binding, and cerebral cortex development (Table 2), suggesting that dasatinib and quercetin could be related to inflammation reduction and neuronal survival and regeneration.

In this study, our SYBR Green PCR analysis demonstrated that the expressions of *Slc38a2* and *Mthfd2* genes were significantly downregulated in the retina with quercetin treatment, whereas the *Metrn* gene was significantly upregulated in the dasatinib-treatment-alone and combined treatment of dasatinib and quercetin groups (Figure 4). The higher expression of meteorin protein in the GCC layer was further confirmed by the immunofluorescence analysis (Figure 5). Slc38a2 expression has been found to be induced in rat brain subjected to prolonged systemic hypertonicity [32] and upregulated in hypothalamic cell line N25/2 subjected to complete amino acid starvation [33], while *Mthfd2*, the regulatory enzyme of mitochondrial folate cycle, was found to be upregulated in mice with methylenetetrahydrofolate reductase deficiency and hyperhomocysteinemia [34] and linked to DNA polymerase gamma expression, a common cause of mitochondrial neurodegeneration [35]. These reflect that the downregulation of *Slc38a2* and *Mthfd2* genes in the quercetin treatment could be related to RGC survival promotion [11]. Meteorin is a secreted protein that regulates glial cell differentiation and promotes axonal extension [36]. It has been reported that lentiviral delivery of meteorin protects striatal neurons against excitotoxicity and reverses motor deficits in rats treated with quinolinic acid [37], and recombinant meteorin can promote the migration of neuroblasts from the subventricular zone explants of postnatal rats and stroke-subjected adult rats and reduce N-methyl-D-aspartate-induced apoptotic cell death of subventricular zone cells in vitro [38]. Yet another study reported that overexpression of *Metrn* did not enhance injury-induced striatal neurogenesis but significantly increased the proportion of new cells with astroglial and oligodendroglial features [39]. The mechanisms behind the involvement of meteorin in the promotion of RGC survival, dendritic repair, and axonal regeneration post-ON injury require further investigation.

There were several limitations in this study. First, a single dose of dasatinib and quercetin, based on the previously reported concentrations [15], was used in this study. Second, only a small number of protein-coding genes were differentially expressed in the retina of mice that received the combined treatment of dasatinib and quercetin, which could be due to the dosage of the treatment. Further investigations on different dosages of dasatinib and quercetin can help to formulate the optimal dosage for RGC survival promotion, dendritic repair, and axonal regeneration. In addition, the whole retina was used in the RNA sequencing analysis. Single-cell analysis can be applied in the future to delineate the differentially expressed genes and their expression specifically in a single RGC.

## 4. Materials and Methods

### 4.1. Animals

The protocols for animal experiments were approved by the Animal Experimentation Ethics Committee of the Joint Shantou International Eye Center of Shantou University and the Chinese University of Hong Kong (approval number: EC20230317(1)-P08) and performed according to the Guidelines of the Association for Research in Vision and Ophthalmology Statement on the Use of Animals in Ophthalmic and Vision Research. Two-month-old male B6.Cg-Tg(Thy1-YFP)HJrs/J (Thy1-YFP-H) mice (Jackson Laboratory, Bar Harbor, ME, USA; for dendrite analyses) and C57BL/6J (Beijing Vital River Laboratory Animal Technology Co., Ltd., Beijing, China) were maintained in a specific pathogen-free grade animal laboratory at 22 ± 1 °C with 40 ± 10% humidity and a 12 h dark/light cycle. Water and standard rodent chow were provided ad libitum. Three mice in each group were used for RNA sequencing analysis, and five mice in each group were used for other experiments.

### 4.2. Optic Nerve Crush Injury

A unilateral ON crush injury was performed in mice based on our previous study [3]. Briefly, the mice were anesthetized with an intramuscular injection (0.1 mL/100 g) of a 1:1 mixture of 20 mg/mL xylazine (Sigma-Aldrich, St. Louis, MO, USA) and 70 mg/mL Zoletil^®^ (Virbac, Carros, France). The right eye received a horizontal incision at the fornical conjunctiva. The ON was then uncovered via blunt separation of the extraocular muscles and adipose tissue using ophthalmic forceps. Under an operating microscope, a crush injury was made at 1.5 mm behind the ON head by an angled jeweler’s forceps (Dumont #5; Roboz, Rockville, MD, USA) for 5 s. The damage to the ophthalmic artery beneath the ON should be avoided. An effective ON crush injury was confirmed by the presence of a clear gap across the entire ON at the crush site.

### 4.3. Treatment of Dasatinib and Quercetin

Dasatinib and quercetin treatments were administered according to our previous study [11]. Briefly, quercetin (Sigma-Aldrich) was dissolved in polyethylene glycol 300 (PEG300), whereas dasatinib (Sigma-Aldrich) was first dissolved in dimethyl sulfoxide (DMSO). Dasatinib and quercetin were then diluted before administration with saline, reaching the solvent concentrations of 1% DMSO and 10% PEG300, respectively. The mice with the ON crush injury were intragastrically fed with 5 mg/kg dasatinib and/or 50 mg/kg quercetin every day for 14 days. In the vehicle control group, the mice were fed with an equal volume of saline with 10% PEG300 and 1% DMSO every day for 14 days. At Days 5 and 14 post-ON injury, the mice were sacrificed for further evaluations.

### 4.4. Retinal Ganglion Cell Dendrite Analysis

Longitudinal in vivo RGC dendrite analysis was performed based on our previous report [40]. In brief, the Thy1-YFP-H mice were anesthetized by intramuscular injection (0.1 mL/10 g) of a 1:1 mixture of Zoletil^®^ (70 mg/mL) and xylazine (20 mg/mL). The fundus of the mice was imaged by a RETImap^®^ machine (Roland, Keltern, Germany) in vivo under the auto-fluorescence module with a blue laser source of 488 nm and a 100 D corneal contact lens before ON injury, and at 4, 7, and 14 days after ON injury. The photographs were taken within a 30° circle surrounding the optic nerve head. Fifteen images were captured at the exact retinal location per second and averaged automatically by the built-in software to augment the signal-to-noise ratio. The dendrites were analyzed using the Fiji software (version 1.54g) [41] with Simple Neurite Tracer [42] and Sholl Analysis plug-ins [43]. In each imaged RGC, dendritic field area (the area bounded by the ends of all terminal dendritic branches), the number of terminal branches (total number of terminal dendritic ends), total dendritic branch length (sum of the lengths of all dendrites), dendritic branching complexity (total number of intersections between RGC filament and concentric circles drawn at 10-μm intervals from the soma in the Sholl analysis), and maximum Sholl score (the number of intersections with the single concentric circle that was the largest of the set for each RGC) were analyzed. For each group, at least 100 RGCs were analyzed.

### 4.5. Retinal Ganglion Cell Axonal Regeneration Analysis

The axonal regeneration analysis was performed based on our previous report [10]. Briefly, the C57BL/6J mice with different treatments were sacrificed at Day 14 post-ON injury and perfused with 4% paraformaldehyde, and the eyeball with attached ON segments was separated from the connective tissues. The dissected ON was further fixed in 4% paraformaldehyde overnight and treated with a 10–30% sucrose gradient. The ON was then cryo-embedded in optimal cutting temperature compound, and longitudinal ON frozen sections (10 μm) were mounted onto the glass slides, washed with tris-buffered saline (TBS), and treated with 0.1% H_2_O_2_ in methanol for 10 min. After 1 h blocking in 5% bovine serum albumin (BSA), the sections were probed with the anti-Gap43 antibody (catalog number: ab75810; Abcam, Cambridge, UK) at 4 °C overnight, washed with Triton X-100- and Tween-20-supplemented TBS (TBS_2_T) at 4 °C for 1 h, and probed again with the anti-Gap43 antibody at room temperature for 1 h. After 1 h TBS_2_T incubation at room temperature, the sections were probed with the secondary antibody conjugated with Alexa Fluor-555 at room temperature for 2 h. The stained ON sections were mounted with the anti-fading mounting medium, and the fluorescence signals were visualized under a confocal microscope (Leica TCS SP5 II; Leica Microsystems, Wetzlar, Germany). The regenerated axons at the distal point of 0.25 and 0.5 mm from the crush site were counted, and the total number of regenerated axons in each section was calculated as previously described [8].

### 4.6. Retinal Transcriptomic Analysis

Retinal transcriptomic analysis was conducted based on our previous report [44]. Dasatinib-, quercetin-, and vehicle-treated C57BL/6J mice were sacrificed at Day 5 post-ON injury. The retinas were dissected, and total RNA was extracted by the TRIzol reagent (Thermo Fisher Scientific, Waltham, MA, USA). RNA sequencing experiments were performed by the Novogene Co., Ltd. (Beijing, China). Briefly, after RNA integrity evaluation, mRNA was isolated from total RNA using oligo dT magnetic beads, and complementary DNA was synthesized. With adenylation at the 3′ ends of DNA fragments, the hairpin loop structure adaptor was ligated for hybridization. The 370–420-base pair (bp) cDNA fragments were amplified by polymerase chain reaction (PCR), purified, and the quality was assessed. The library preparations were sequenced using the Illumina Novaseq 6000 platform (Illumina, San Diego, CA, USA) to generate 150 bp paired-end reads. The low-quality reads and the reads containing the adapter or poly-N were removed from the raw data. The paired-end clean reads were aligned to the reference genome by Hisat2 v2.0.5. The number of reads mapped to each gene was counted by FeatureCounts v1.5.0-p3, and the fragments per kilobase of transcript sequence per million base pairs sequenced (FPKM) of each gene was calculated. Principal component analysis (PCA) on whole RNA sequencing profiles was used to confirm the distinct clustering of different treatment groups. Differential gene expression analysis was performed using the DESeq2 R package (1.20.0). Differential gene expression was considered as log_2_ fold change ≥1 or ≤−1 and adjusted *p* (*p_adj_*) < 0.05 (Benjamini & Hochberg method). A volcano plot was used to display the differentially expressed genes in the retinas of the mice that received the combined treatment of dasatinib and quercetin, as compared to those of the vehicle treatment. Hierarchical clustering analysis was also adopted to visualize the distinct patterns of differentially expressed genes in different groups. The biological functions of the differential expressed genes were evaluated by the gene ontology analysis (DAVID Bioinformatics Resources 6.8). *p* < 0.05 was considered statistically significant.

### 4.7. SYBR Green PCR Validation Analysis

All 15 differentially expressed protein-coding genes in the RNA sequencing analysis (*Fat4*, *Fosl2*, *Fstl5*, *Metrn*, *Mthfd2*, *Nrtn*, *Papolb*, *Pcsk1*, *Pdc*, *Rpl19*, *Ryr3*, *Slc38a2*, *Tma7*, *Uba6*, and *Zfhx4*) were subjected to further SYBR green PCR validation in mice with individual and combined treatments of dasatinib and quercetin. Complementary DNA was synthesized from total RNA of mouse retina by SuperScript III reverse transcriptase (Thermo Fisher Scientific) and amplified with SYBR Green I Master Mix (TaKaRa Bio Inc., Shiga, Japan) and respective specific primers (Appendix A) processed in the LightCycler 480 system (Roche, Basel, Switzerland). Relative expression was calculated by the 2^−ΔΔCt^ method with normalization by *Actb* housekeeping gene and compared with the vehicle treatment group.

### 4.8. Immunofluorescence Analysis

At Day 5 post-ON injury, the C57BL/6J mice with the different treatments were sacrificed and perfused with 4% paraformaldehyde. The eyeballs were enucleated for post-fixation in 4% paraformaldehyde at 4 °C overnight. The fixed eyeballs were cryo-protected with a 10–30% sucrose gradient and cryo-embedded in optimal cutting temperature compound. The cryo-sections (10 μm) with pupil–ON position were blocked and permeabilized in PBS with 5% NGS and 0.3% Triton X-100 at room temperature for 1 h, probed with primary antibody against meteorin (catalog no.: PA5-121777; Invitrogen, Carlsbad, CA, USA) at 4 °C overnight, and incubated with Alexa Fluor-488-conjugated secondary antibody at room temperature for 2 h. The stained sections were mounted and imaged by a confocal microscope (Leica TCS SP5 II). Twenty retinal images at 200 μm from the ON were analyzed for each group. The fluorescence signal intensity in the GCC layer was measured by the ImageJ software (version 1.54g; National Institutes of Health, Bethesda, MD, USA) in each retinal section.

### 4.9. Statistical Analysis

Data were presented as means of the results from five mice ± standard deviation (SD) and analyzed by one-way analysis of variance (ANOVA) with post hoc Fisher’s least significant difference (LSD) test. The statistical analyses were performed by IBM SPSS Statistics 26 (SPSS Inc., Chicago, IL, USA). *p* < 0.05 was considered statistically significant.

## 5. Conclusions

This study revealed that individual and combined treatments of dasatinib and quercetin reduced RGC dendritic shrinkage and promoted axonal regeneration in mice after ON injury, probably mediated through meteorin. Our results suggest the treatment potentials of dasatinib and quercetin for RGC repair and regeneration in traumatic optic neuropathy.

## Figures and Tables

**Figure 1 ijms-26-12170-f001:**
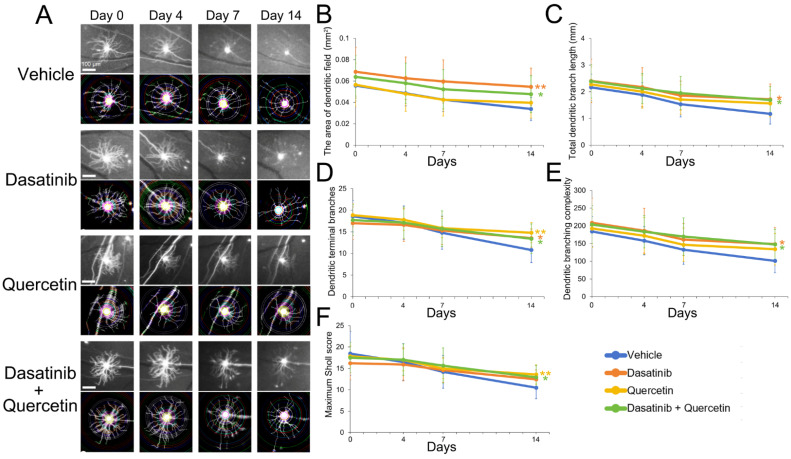
Dasatinib and quercetin treatments alleviate retinal ganglion cell dendritic shrinkage in mice post-optic nerve injury. (**A**) Confocal scanning laser ophthalmoscope images and dendritic skeleton images of retinal ganglion cell (RGC) dendrites of mice with dasatinib and quercetin treatments longitudinally from Day 0–14 post-optic nerve injury. Scale bar—200 μm. Colored circle lines—concentric circles drawn at 10-μm intervals from the soma in the Sholl analysis. (**B**–**F**) Quantitative analysis on (**B**) the area of dendrite field, (**C**) total dendritic branch length, (**D**) dendritic terminal branches, (**E**) dendritic branching complexity, and (**F**) maximum Sholl score compared to those in vehicle treatment. Data were presented as mean ± standard deviation and compared by one-way analysis of variance with post hoc LSD test. * *p* < 0.05; ** *p* < 0.01 as compared to the mice with vehicle treatment.

**Figure 2 ijms-26-12170-f002:**
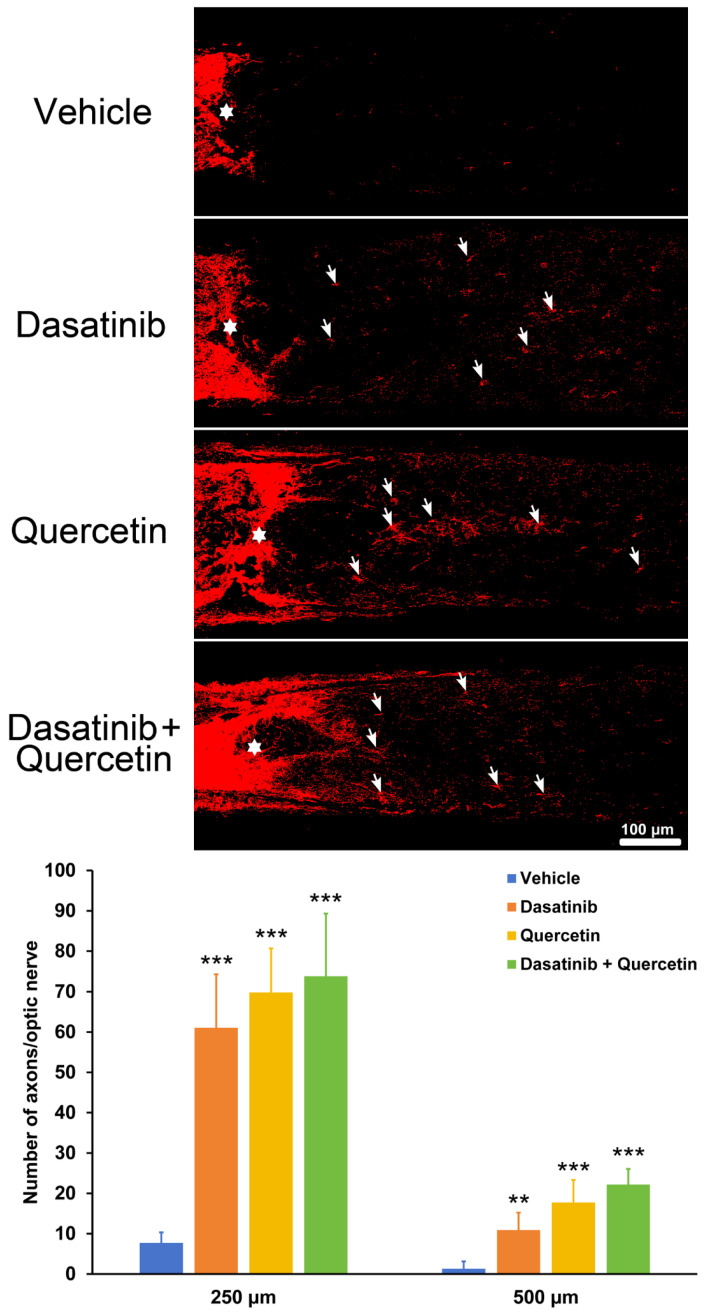
Dasatinib and quercetin treatments promote retinal ganglion cell axonal regeneration in mice post-optic nerve injury. Immunofluorescence analysis of Gap43 on the optic nerve of mice with dasatinib and quercetin treatments at Day 14 post-optic nerve injury and quantification of the regenerated axons (arrows) at 250 and 500 μm from the crush site (asterisks). Scale bar—100 μm. Data were presented as mean ± standard deviation and compared by one-way analysis of variance with post hoc LSD test. ** *p* < 0.01; *** *p* < 0.001 as compared to the mice with vehicle treatment.

**Figure 3 ijms-26-12170-f003:**
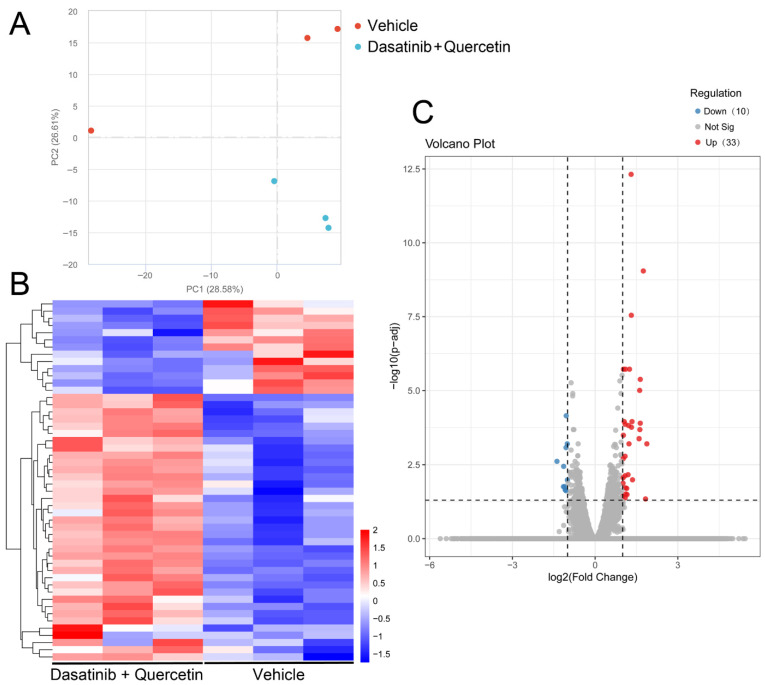
RNA sequencing analysis on the retina of mice with the combined treatment of dasatinib and quercetin post-optic nerve injury. (**A**) Principal component analysis of the whole RNA sequencing profiles of the retinas of mice with the vehicle treatment (red dots) and the combined treatment of dasatinib and quercetin (blue dots) at Day 5 post-optic nerve injury. (**B**) Hierarchical clustering analysis of the differentially expressed genes in the retinas of mice with the vehicle treatment and the combined treatment of dasatinib and quercetin. (**C**) Volcano plots of the gene expression changes in the retinas of mice with the combined treatment of dasatinib and quercetin, as compared to those with the vehicle treatment, by an independent *t*-test. Red dots—genes with significant upregulation; blue dots—genes with significant downregulation; gray dots—genes with no significant changes. Differential expression was defined as log_2_ fold change ≥ 1 and adjusted *p* < 0.05 (dashed lines).

**Figure 4 ijms-26-12170-f004:**
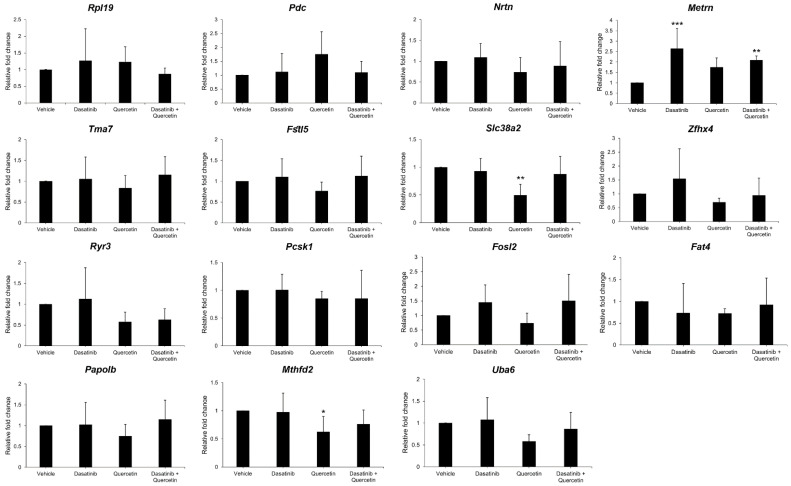
Gene expression analysis on the retinas of mice with individual or combined treatments of dasatinib and quercetin post-optic nerve injury. SYBR green polymerase chain reaction on the expression of all 15 differentially expressed protein-coding genes identified by the RNA sequencing analysis (*Fat4*, *Fosl2*, *Fstl5*, *Metrn*, *Mthfd2*, *Nrtn*, *Papolb*, *Pcsk1*, *Pdc*, *Rpl19*, *Ryr3*, *Slc38a2*, *Tma7*, *Uba6*, and *Zfhx4*) in the retina of mice with individual or combined treatments of dasatinib and quercetin at Day 5 post-optic nerve injury, as compared to that with vehicle treatment. *Actb* was adopted as the housekeeping gene for normalization. Data presented as mean of relative fold changes (2^−ΔΔCt^) ± standard deviation and compared by one-way analysis of variance with post hoc LSD test. * *p* < 0.05; ** *p* < 0.01; *** *p* < 0.001.

**Figure 5 ijms-26-12170-f005:**
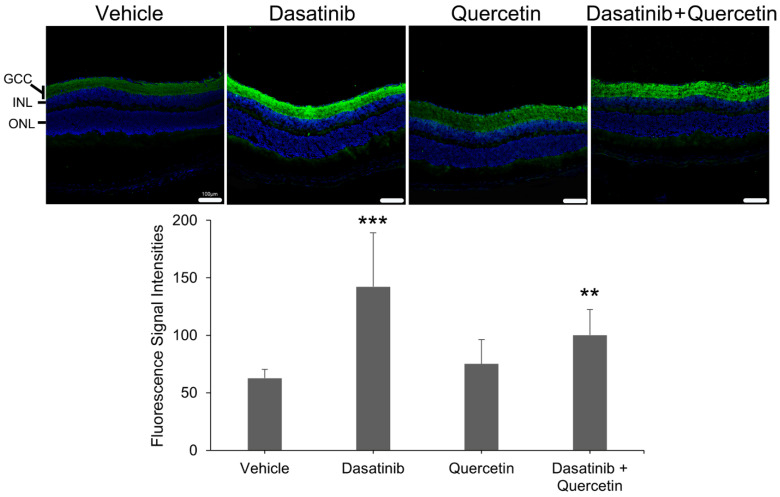
Meteorin protein expression in the ganglion cell complex layer of mice with individual or combined treatments of dasatinib and quercetin post-optic nerve injury. Immunofluorescence analysis on meteorin protein expression in the retinal sections of mice with individual or combined treatments of dasatinib and quercetin at Day 5 post-optic nerve injury and quantification of the immunofluorescence signal intensities of meteorin in the ganglion cell complex (GCC) layer. Data were presented as mean ± standard deviation and compared by one-way analysis of variance with post hoc LSD test. Scale bar—100 μm. Green—meteorin signal; blue—DAPI nuclei counter-stain; INL—inner nuclear layer; ONL—outer nuclear layer. ** *p* < 0.01; *** *p* < 0.001 as compared to the mice with vehicle treatment.

**Table 1 ijms-26-12170-t001:** Differentially expressed protein-coding genes in the retina of mice with the combined treatment of dasatinib and quercetin at Day 5 post-optic nerve injury.

Genes	Average FPKM (Dasatinib and Quercetin)	Average FPKM (Vehicle)	log_2_ Fold Change	*p_adj_*
*Rpl19*	131.89	42.88	1.62	2.18 × 10^−4^
*Pdc*	86,158.36	34,833.44	1.31	4.84 × 10^−13^
*Nrtn*	471.31	201.29	1.23	6.27 × 10^−4^
*Metrn*	116.95	55.10	1.09	0.042
*Tma7*	2652.60	1300.35	1.03	1.86 × 10^−6^
*Fstl5*	432.41	868.10	−1.01	0.010
*Uba6*	162.91	327.84	−1.01	6.34 × 10^−4^
*Slc38a2*	817.78	1690.88	−1.05	6.93 × 10^−5^
*Zfhx4*	201.65	418.33	−1.05	8.45 × 10^−4^
*Ryr3*	67.32	139.93	−1.06	0.024
*Pcsk1*	90.03	190.38	−1.09	0.022
*Fosl2*	118.21	255.20	−1.11	0.018
*Fat4*	87.83	193.34	−1.14	0.004
*Papolb*	50.46	111.78	−1.15	0.018
*Mthfd2*	155.78	406.21	−1.38	0.002

FPKM—fragments per kilobase of transcript per million fragments; *p_adj_*—adjusted *p*-value.

**Table 2 ijms-26-12170-t002:** Gene ontology analysis on the differentially expressed protein-coding genes in the retinas of mice with the combined treatment of dasatinib and quercetin at Day 5 post-optic nerve injury.

Functional Annotations	Genes	%	*p*
Response to glucocorticoid	*Pcsk1*, *Fosl2*	13.33	0.027
Calcium ion binding	*Fstl5*, *Fat4*, *Ryr3*	20.00	0.041
Cerebral cortex development	*Fat4*, *Slc38a2*	13.33	0.043

## Data Availability

The original contributions presented in this study are included in the article/Appendix A. Further inquiries can be directed to the corresponding author.

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
