# Peer review of "Dasatinib and Quercetin Alleviate Retinal Ganglion Cell Dendritic Shrinkage and Promote Axonal Regeneration in Mice with Optic Nerve Injury"

_ijms, 2025, doi:10.3390/ijms262412170_

Round 1

Reviewer 1 Report

Comments and Suggestions for Authors

There were several studies regarding the senolytic therapy with Dasatinib and Quercetin including these authors' previous publications. The novelty of the treatments is not high and the authors emphasized "for the first time, identified that individual and combined treatment of dasatinib and quercetin can alleviate RGC dendritic shrinkage and promote axonal regeneration in mice after ON injury (Figure 1 and 2). There are some suggestions that the authors should add/modify in the method and results. First, I did not see the optic nerve crush in the methods. The figure 2 GAP43 immunostained optic nerve images did not show significantly improvement compared with the vehicle. The vehicle treated optic nerve images (top in the figure 2) still show some axons on the right side. This makes readers think that the optic nerve crush might not generate the same damages in all mice. Did the authors use the Cholera toxin B subunit (CTB) intravitreal injection to trace how far it can travel through the optic nerve in vivo? In the figure 3A, please explain why the vehicle treated data has wild spread three red dots. Please address above points and provide the minor revision.

Author Response

Dear Reviewer 1,

Thank you for your time to review our manuscript! We greatly appreciate your comments and suggestions. We have revised the manuscript accordingly with yellow highlights. Thank you very much again for your kind consideration of our manuscript in International Journal of Molecular Sciences.

Yours sincerely,

Tsz Kin Ng

Reviewer 1

  1. I did not see the optic nerve crush in the methods.

Response: We apologize for the missing information. The methods for the optic nerve crush injury has been added to the Materials and Methods section of the revised manuscript (Page 10: Line 306-316. 4.2. Optic nerve crush injury).

  1. The figure 2 GAP43 immunostained optic nerve images did not show significantly improvement compared with the vehicle. The vehicle treated optic nerve images (top in the figure 2) still show some axons on the right side. This makes readers think that the optic nerve crush might not generate the same damages in all mice. Did the authors use the Cholera toxin B subunit (CTB) intravitreal injection to trace how far it can travel through the optic nerve in vivo?.

Response: We apologize for the confusion. Figure 2 have been replaced by better quality images.

  1. In the figure 3A, please explain why the vehicle treated data has wild spread three red dots.

Response: Thank you for the question. Three separate retina samples in each group were used in the RNA sequencing analysis. We admit that the transcriptome profile of one sample in the vehicle control group was a bit far away from 2 other samples; yet, overall the transcriptome profiles of the 3 samples in the vehicle control group were distinctly separate from that of the combined dasatinib and quercetin treatment group.

Reviewer 2 Report

Comments and Suggestions for Authors

This is an interesting manuscript that is aiming to present role of senolytics (dasatinib and quercetin) treatments can promote RGC survival and electroretinography activity. Their prior paper presented role of senolytics (dasatinib and quercetin) treatments can promote RGC survival and alleviate the reduction of ganglion cell complex thickness. Authors need to carefully check and ensure that their prior paper (PMID: 39021340 ) and this manuscript do not have overlaps. Why the Figure 3 of this manuscript bears striking resemblance to the Figure 1 of their prior paper (PMID: 39021340)? It is good to edit the paper and present non overlapping data and describe how exactly this manuscript complements prior paper. In their cover letter authors should explain why they split the data to present in two different manuscripts. As presented this manuscript has some unacceptable overlap with the prior published paper. However, they have some non-overlapping data. Striking resemblance of the figures as noted above need to be explained. 

Comments on the Quality of English Language

As noted in comments above, authors need to improve presentation. 

Author Response

Dear Reviewer 2,

Thank you for your time to review our manuscript! We greatly appreciate your comments and suggestions. We have revised the manuscript accordingly with yellow highlights. Thank you very much again for your kind consideration of our manuscript in International Journal of Molecular Sciences.

Yours sincerely,

Tsz Kin Ng

Reviewer 2

  1. Authors need to carefully check and ensure that their prior paper (PMID: 39021340 ) and this manuscript do not have overlaps. Why the Figure 3 of this manuscript bears striking resemblance to the Figure 1 of their prior paper (PMID: 39021340)? It is good to edit the paper and present non overlapping data and describe how exactly this manuscript complements prior paper.

Response: Thank you for the comment and question. Although both Figure 3 in this study and Figure 1 from our previous study (Yao et al., 2024; PMID: 39021340) presented the results of RNA sequencing analysis, Figure 1 from our previous study (Yao et al., 2024; PMID: 39021340) presented the RNA sequencing results of the rat retinas at Day 7 post-optic nerve crush injury and the rat retinas without optic nerve crush injury, whereas Figure 3 in this study presented the RNA sequencing results of the retinas of mice with combined dasatinib and quercetin treatment at Day 5 post-optic nerve crush injury and the retinas of mice with vehicle treatment at Day 5 post-optic nerve crush injury.

  1. In their cover letter authors should explain why they split the data to present in two different manuscripts.

Response: Thank you for the question. The results of post-optic nerve injury retinal ganglion cell survival with dasatinib and quercetin treatments in our previous study (Yao et al., 2024; PMID: 39021340) aimed to confirm the involvement of cellular senescence in retinal ganglion cell death regulation after optic nerve crush injury and to verify the effect of transforming growth factor-β receptor type II inhibitor (LY2109761) treatment on retinal ganglion cell survival regulation. After the completion of that study, we believe that, in addition to retinal ganglion cell survival, we can further investigate the effect of dasatinib and quercetin treatments on retinal ganglion cell dendrites and axons; therefore, the experimental results of dasatinib and quercetin treatments on retinal ganglion cell dendrites and axons were presented in this study.

Round 2

Reviewer 2 Report

Comments and Suggestions for Authors

Authors should add summary of response to point 2, " The results of post-optic nerve injury retinal ganglion cell survival with dasatinib and quercetin treatments in our previous study (Yao et al., 2024; PMID: 39021340) aimed to confirm the involvement of cellular senescence in retinal ganglion cell death regulation after optic nerve crush injury and to verify the effect of transforming growth factor-β receptor type II inhibitor (LY2109761) treatment on retinal ganglion cell survival regulation. After the completion of that study, we believe that, in addition to retinal ganglion cell survival, we can further investigate the effect of dasatinib and quercetin treatments on retinal ganglion cell dendrites and axons; therefore, the experimental results of dasatinib and quercetin treatments on retinal ganglion cell dendrites and axons were presented in this study."  as they deem appropriate in the introduction and also in the discussion in order to add clarity that it is not a duplicate publication. 

Comments on the Quality of English Language

As noted in comments above, authors need to improve presentation. 

Author Response

Dear Reviewer 2,

Thank you for your time again to review our revised manuscript! We greatly appreciate your comments and suggestions. We have revised the manuscript accordingly with yellow highlights. Thank you very much for your kind consideration of our manuscript in International Journal of Molecular Sciences.

Yours sincerely,

Tsz Kin Ng

Reviewer 2

  1. Authors should add summary of response to point 2, " The results of post-optic nerve injury retinal ganglion cell survival with dasatinib and quercetin treatments in our previous study (Yao et al., 2024; PMID: 39021340) aimed to confirm the involvement of cellular senescence in retinal ganglion cell death regulation after optic nerve crush injury and to verify the effect of transforming growth factor-β receptor type II inhibitor (LY2109761) treatment on retinal ganglion cell survival regulation. After the completion of that study, we believe that, in addition to retinal ganglion cell survival, we can further investigate the effect of dasatinib and quercetin treatments on retinal ganglion cell dendrites and axons; therefore, the experimental results of dasatinib and quercetin treatments on retinal ganglion cell dendrites and axons were presented in this study." as they deem appropriate in the introduction and also in the discussion in order to add clarity that it is not a duplicate publication.

Response: Thank you for the comments. The corresponding information has been added to the Introduction and Discussion of the revised manuscript (highlighted in manuscript).